# Detecting Traces of 17α-Ethinylestradiol in Complex Water Matrices

**DOI:** 10.3390/s20247324

**Published:** 2020-12-20

**Authors:** Paulo M Zagalo, Paulo A Ribeiro, Maria Raposo

**Affiliations:** CEFITEC, Departamento de Física, Faculdade de Ciências e Tecnologia, Universidade Nova de Lisboa, 2829-516 Caparica, Portugal; p.zagalo@campus.fct.unl.pt

**Keywords:** EE2, hormone, impedance, capacitance, loss tangent, electric modulus, sensor, interdigitated electrodes, PCA, layer-by-layer films

## Abstract

Hormones have a harmful impact on the environment and their detection in water bodies is an urgent matter. In this work, we present and analyze a sensor device able to detect traces of the synthetic hormone 17α-ethinylestradiol (EE2) below 10^−9^ M in media of different complexities, namely, ultrapure, mineral and tap waters. This device consists of solid supports with interdigitated electrodes without and with a polyethylenimine (PEI) and poly (sodium 4-styrenesulfonate) (PSS) layer-by-layer film deposited on it. Device response was evaluated through capacitance, loss tangent and electric modulus spectra and the data were analyzed by principal component analysis method. While the three types of spectra were demonstrated to be able to clearly discriminate the different media, loss tangent spectra allow for the detection of EE2 concentration, with a sensitivity of −0.072 ± 0.009 and −0.44 ± 0.03 per decade of concentration, for mineral and tap water, respectively. Detection limits values were found to be lower than the ones present in the literature and presenting values of 8.6 fM (2.6 pg/L) and of 7.5 fM (22.2 pg/L) for tap and mineral waters, respectively. Moreover, the obtained response values follow the same behavior with EE2 concentration in any medium, meaning that loss tangent spectra allow the quantification of EE2 concentration in aqueous complex matrices.

## 1. Introduction

17α-ethinylestradiol (EE2) is a man-made synthetic female hormone with a rather high estrogenic action that, even at extremely low concentrations, has a strong biological impact, approximately 10 times higher than estrogens naturally produced by the human body [1,2]. This compound is commonly used in treatments aimed at osteoporosis, hormone imbalance and infertility issues and is one of the main components present in female birth control pills [3]. Due to its main uses, this hormone is a pharmaceutical and personal care product (PPCP) and, given its strong biological effects and consequences, it is also labeled as one of the more predominant and active endocrine disruptors (EDC) present in the environment [4]. When faced with environmental risks brought on by compounds such as this, relevant policies and legislations must be passed and used to create watch lists, limit usage and/or even ban certain molecules in order to preserve the environment and population health quality in general.

17α-ethinylestradiol, as well its non-synthetic estrogen counterparts, such as estrone (E1) and 17β-estradiol (E2), eventually end up entering wastewater treatment plants (WWTPs) where, together with other compounds, they are subjected to physical–chemical and biological treatments in order to be removed from the water, which, in turn, is discharged onto rivers and other water bodies. However, the removal of EE2, as well as of other compounds, is not flawless, thus resulting in its presence in the environment to the point of surpassing levels of predicted no-effect concentrations (PNEC) regarding ecological toxicity [5]. Even though this hormone can be considered somewhat stable, its widespread usage and ceaseless build-up in the environment, namely in lakes, ponds, muds, rivers and underground waters have led to several documented cases of toxic and harmful consequences on both fauna and flora life cycles [6,7].

Some examples of these repercussions consist of physiological and morphological variations in cows, which, in turn, bloom into changes in teat length and in the hue of the vulva; a lower sperm count, fluctuations in the reproductive fitness and alterations in other reproductive properties in fish populations due to the feminization of male fishes; some cases of uterine prolapse in pigs; the possibility of the origin of abnormal behaviors and reactions in sheep and cows such as ovarian cysts, nymphomania or even premature development of the udders; a herd of sheep was rendered permanently infertile upon feeding on cloves present in a field that was irrigated with water containing unnatural concentration levels of estrogen; using treated wastewater to irrigate crop fields resulted in the decreased fertility of planted vegetables; an increased probability in humans to develop severe medical conditions, namely breast or prostate cancer, or even cardiovascular diseases [8,9,10,11,12]. Due to the imposing risk and pernicious consequences to the environment, this issue has been analyzed and discussed, giving way to the creation of policies and regulations such as the European Union Decision 840 of 5 June 2018, that aims to place EE2, and other similar composites, on strict watch lists that may, in the future, limit or even ban its usage [13]. In recent years, studies have been conducted to delve deeper into this subject and compare the impact of EE2 in different regions around the world [14]. Moreover, such studies demonstrate that the detection and monitoring of hormones in different water bodies are a priority.

Different types of EE2 sensors can be found in the literature [15,16,17,18], however, measuring techniques based in electrochemistry have limitations when the sensors are intended to be easy to handle, thus not relying on the presence and skill of specialized operators. For example, amperometry and cyclic voltammetry operates in complex liquids, requiring compounds that can be oxidized or reduced actively on the working electrode which, in turn, requires these techniques to be overseen and conducted by a specialized operator.

The electronic tongue (ET) system concept, based on an array of sensorial units, such as [19] and references therein, is an interesting option when it comes to detecting hormones in aqueous complex media. Nonetheless, it will be necessary to find innovative sensors, which are easy to use and have better features than those that already exist, while simultaneously undertaking the quest to achieve efficient and reliable sensors. The use of impedance spectroscopy such as the electrical characterization technique makes the ET system versatile and allows us to study the electrical properties of water samples and infer their behavior and response to a varied set of stimuli. However, an ET system also requires the development of adequate sensorial units to detect a determined molecule or molecular group. Sensorial units can be prepared by depositing thin films onto solid supports with interdigitated electrodes (IDE). The electrical measurements are performed while immersing the sensorial units into the aqueous media that are under analysis. Therefore, the electrical measurements are dependent on the electrical characteristics of the deposited thin film, and the double layer is formed when the thin film is immersed in the liquid and the liquid bulk itself. The equivalent electrical circuit, which represents the electrical behavior of IDE covered with a thin film immersed in aqueous solutions, was proposed by Taylor and Macdonald [20] and, recently, was revisited by Elamine et al. [21]. Furthermore, subsequent tests following the abovementioned Taylor and Macdonald model showed that it is possible to identify conductive and capacitive proprieties throughout the measurement of capacitance and loss tangent spectra in a range of frequencies, as well as demonstrating that the measured electrical behavior can distinguish different aqueous media. Nonetheless, although the capacitance spectra are commonly used as a tool to study and monitor the electrical response of interdigitated sensors [22,23], the use of loss tangent spectra is not commonly used. However, these tangent spectra seem to be rather interesting since the measured values are dependent on the electrical properties of the materials and independent of the device dimensions. This then becomes a valuable tool when it comes to data comparison and to avoid value discrepancies when, for example, the IDE are not completely immersed in the liquid.

Impedance data also allow us to calculate the electric modulus spectra, whose imaginary part manages to suppress highly capacitive phenomena, thus avoiding features of the spectra from being masked by space charge effects [24,25], and is usually associated with protonic conduction [26,27].

In the present work, we analyzed the effect of the use of capacitance, loss tangent and imaginary electric modulus spectra on the results of an electronic tongue based on a simple array of sensorial units to detect EE2 in aqueous solutions with different complexities. The sensorial units were solid supports with interdigitated gold electrodes deposited on one of the surfaces and were used without any thin film coverage and with polyethylenimine (PEI) and poly (sodium 4-styrenesulfonate) (PSS) layer-by-layer (LbL) thin films [28]. The electrical spectra were analyzed by the principal component analysis method (PCA) [29]. The results demonstrated that the PCA analysis of capacitance, loss tangent and electric modulus spectra give similar conclusions: the samples can be discriminated by type of water used in the preparation of samples. Moreover, when the loss tangent spectra of MW and TW are analyzed and the principal component 2 (PC2) values are compared, a constant proportionality is observed between PC2 values for EE2 concentrations, demonstrating that the present ET can also present quantitative features.

## 2. Materials and Methods

Ceramic substrates with deposited gold IDE and 200 μm digit width with 200 μm distance between digits were acquired from Metrohm, Spain. These solid supports with IDE, uncoated and coated with a thin film, were used as sensing devices to detect 17α-ethinylestradiol (EE2) in three different water matrices with distinct complexity levels. The water samples used were ultrapure water (UW) (pH = 6.5 ± 0.3), a commercial Portuguese mineral water (MW) (pH = 5.7 ± 0.3) and tap water (TW) (pH = 6.8 ± 0.1). These aqueous matrices were chosen in order to observe how differing water complexities would influence the electrical measurements as well as the hormone detection. The ultrapure water was obtained in a Milli-Q ultrapure water system (Millipore GmbH, Billerica, MA, USA), mineral water came from a commercial Portuguese mineral water brand and tap water was collected at Faculdade de Ciências e Tecnologia (Caparica, Portugal). Regarding the sensorial devices, two distinct types of sensors were prepared: uncoated IDE and IDE coated with a thin film. The thin films deposited onto the IDE were attained through the layer-by-layer (LbL) technique, associated with adsorption by electrostatic forces [28] or other physical interactions [30,31,32], which consists of the alternate deposition of layers of polyelectrolytes with opposing electrical charges in order to achieve several bilayers. These types of films have been demonstrated to be useful in the development of sensors in previous works [33,34]. The negative polyelectrolyte used was polyethylenimine (PEI), and its positively charged counterpart was poly (sodium 4-styrenesulfonate) (PSS), prepared using aqueous solutions with a $10^{-2}\;$M concentration of both polyelectrolytes. The adsorption time of each polyelectrolyte layer was of 30 s. After the adsorption of each bilayer, the film was dried gently with a nitrogen flux. It should be noted that this drying process reduces the ionic conductivity of the LbL films, since the nanocrystals formed from the polyelectrolytes counterions are removed from the film after immersion in the next immersing solution [35,36] Therefore, by employing the LbL technique, thin films of PEI/PSS with five bilayers were produced, designated by (PEI/PSS)_5_. These polyelectrolytes were chosen as viable materials in the development of the thin films due to previous results attained for the detection of another PPCP molecule, triclosan (TCS, 2,4,4′-Trichloro-2′-hydroxydiphenyl ether) [37,38].

To characterize the sensorial units’ response when exposed to the hormone, various solutions were prepared with different concentrations of EE2, ranging from the lowest (0 M) to the highest (10^−9^ M), and from different water matrices with increasing complexity, as stated previously, ultrapure, mineral and tap waters. Figure 1 depicts a schematic representation of the experimental setup used in this work.

The electrical measurements were conducted with both type of sensorial units immersed in EE2 solutions and by measuring the impedance spectra at the IDE terminals using a Solartron 1260 Impedance Analyzer, in a frequency range of 1 Hz to 1 MHz and an AC signal voltage of 25 mV. These measurements were performed in triplicate to ensure that the sensors were reproducible. The electrical impedance spectra data features were treated by the Principal Component Analysis (PCA) method in order to reduce the data size and to obtain a new space of orthogonal components to verify if different concentration patterns, clusters and/or order can be observed [29]. PCA should be used in a large number of samples using a large number of variables [39], which is in accordance with the main concept of electronic tongue, where an array of sensors is used to find a large number of variables and compare the measured data with the data of a large number of samples existing in a library. However, from the point of view of sensor development, in which it is necessary to study different proprieties, it becomes impracticable to obtain as many samples as necessary to reduce the error. In this work, one intends to show the proof of concept that the choice of physical quantities may have importance in the sensor response as well as these sensors allowing for quantification in addition to presenting the well-known qualitative character. Therefore, we do not yet envision error reduction and PCA, even though a small number of samples can help, see, for example, [40]. The data used for the PCA analysis were the capacitance, loss tangent and electric modulus spectra collected in a frequency range of 1 Hz to 1 MHz. These spectra were independently used. Each spectrum is the average of three measurements. Each EE2 concentration was characterized by 64 variables, i.e., 32 variables per measurement IEs × 2 types of IDE, and consequently, these variables were organized in database files defined by (1) 24 × 64 (EE2 samples × variables) rectangular matrix in the case of consider ultrapure, mineral and tap waters, and (2) 16 × 64 (EE2 samples × variables) rectangular matrix in the case of consider only mineral and tap waters.

## 3. Results

Figure 2, Figure 3 and Figure 4 show, respectively, the capacitance, loss tangent and electric modulus spectra measured when IDE sensors, uncoated and coated with (PEI/PSS)_5_ thin films, are immersed in solutions with different EE2 concentrations. The solutions were prepared with different aqueous matrices, namely spectra measured in aqueous media UW (a and b), MW (c and d) and TW (e and f) analyzed with uncoated IDE (a, c, e) and IDE covered with (PEI/PSS)_5_ thin-film (b, d, f) IDE sensors. The error bar measured values in these spectra, taking into account the three measurements performed are dependent on frequency and are lower than 1%. These error bars were not included in the spectra of Figure 2, Figure 3 and Figure 4 for better graphic clarity.

When analyzing the spectra displayed in both Figure 2, Figure 3 and Figure 4, it is possible to observe that, in the case of the ultrapure water matrix, the changes in the capacitance, loss tangent loss and electric modulus spectra cannot be correlated with the increasing concentration. Similar results have already been achieved during triclosan detection on ultrapure water, see [33]. However, as the water matrix becomes more complex, although the electric modulus spectra, see Figure 4, do not completely follow the EE2 concentration, capacitance and loss tangent curves become more coherent as the EE2 concentration increases, i.e., there is an evolution towards an ordered sequence between the curves for each concentration. This can be visualized in the insets of Figure 2, Figure 3 and Figure 4. The evolution of the spectra with EE2 becomes clear in Figure 2f and Figure 3f when the sensor coated with a (PEI/PSS)_5_ film is immersed in the most complex water (tap water). The presence of a thin film on the IDE is fundamental to avoid damage caused by the applied electric field to the electrodes during its immersion in the complex aqueous solutions [33]. Consequently, the effect of metallic ions existing in the tap water is reduced. The presence of a thin film with a layer of negative molecules also avoids the adsorption of EE2 on the thin film. This is justified by the ionized form of this molecule, which is anionic, since its pKa value is of about 10 [4]. Consequently, the use of (PEI/PSS)_5_ thin films extends the sensor lifetime.

Repeated measurements also revealed that capacitance, tangent-loss and electric modulus curves are more reproducible in more complex solutions. Moreover, the loss tangent spectra are rather interesting since the measured values evolve more clearly with the concentration of EE2.

To further this comparison, Principal Component Analysis (PCA) of capacitance, loss tangent and electric modulus spectra measured with both types of sensors, when immersed in the solutions of UW, MW and TW, with different EE2 concentration presences, are depicted in the plots of Figure 5.

The PCA plots can provide valuable information regarding the sensors’ capabilities to discriminate each EE2 concentration and to infer the possibility of the eventual occurrence of clusters and whether there are any particular patterns or tendencies to be observed. In this case, the sensor’s ability to discriminate and classify the different EE2-doped water matrices is clearly demonstrated by the PCA plots of Figure 5, obtained from measurements carried out in all three types of water. However, based on the capacitance, loss tangent and elastic modulus spectra present in Figure 2, Figure 3 and Figure 4, the need arose for side-by-side comparison of the behavior of the more complex matrixes, MW and TW, without the influence of UW, which yielded inconclusive results that could not be used in a comparison with the other waters, as illustrated in Figure 6, in which data from both uncoated and coated sensors were used. The comparison of these figures allows us to conclude that although both MW and TW waters are located in separated PCA plot regions, the PCA plot calculated from the different spectra can sort out the MW samples in an ordered sequence of concentration, albeit with less sensitivity than for TW, which not only reveals an ordered sequence from the lowest to the highest concentration but also displays a good separation between concentrations. The PCA plots achieved from loss tangent spectra clearly distinguish the type of water and also the samples by EE2 concentration values. In all spectra types, the spectra analysis displays a higher sensitivity to EE2 concentration and a better signal response when the samples are prepared with TW than when prepared with MW, which is the most complex and rich matrix in the ions of the three that are studied in this work.

## 4. Discussion

Upon close analysis of the spectra in Figure 2, Figure 3 and Figure 4, as the solutions evolve from a simple matrix, UW, to being more complex and “crowded” with ions and other molecules (TW), a right shift takes place in the curves, which is consistent with an increase in the conductivity of the medium which translates into a stronger response at higher frequencies. It is also relevant to note that the measurements conducted for UW do not allow for an ordered distinction between the different concentrations while, for MW, there is a clear improvement, albeit still with poor sensitivity regarding the EE2 concentration level. However, for TW it is possible to observe that not only is there a more ordered sequence from lowest (0 M) to highest (10^−9^ M) EE2 concentration, as illustrated by the red arrow in Figure 3f, there is also a clear separation among them (higher sensitivity), which was further explored in Figure 5 plots. It should be noted here that the presence of a thin film coating on the IDE prevents certain electrochemical reactions that can occur during the electrical measurement. In certain cases, with increasing measuring times, these reactions can lead to a partial removal of the gold electrodes [34]. The presence of a negative polyelectrolyte layer on the sensor’s surface does not allow EE2 or other negative molecules or negative ions to be adsorbed on the surface. This creates a new dynamic in the electric double layer, a distribution in the space of negative and positive ions that appears when a surface is exposed in the aqueous solution and allows the sensor to be used several times. Finally, the more complex a water matrix, the more electrical charges are at play due to the presence of a larger number of ions as well as a higher number of molecules, which allows for higher electrical conduction. It should also be noted that in the case of ultrapure water matrix, the number of ions is reduced, thus making it difficult for the electric double layer to be formed when the sensor is immersed in the water. The random changes present in the impedance spectra when the aqueous matrix is ultrapure water can be explained by the formation of electric double layers, which display a random nature due to the presence of a random number of positive and negative of ions. Concerning the (PEI/PSS)_5_ thin film, PEI (positively charged) and PSS (negatively charged) are both polyelectrolytes and also insulators. Its presence works as capacitor, influencing the features of the impedance spectra measured. 

Building upon what is stated in the previous section, in Figure 5a,b, as highlighted by the colored visual aids, there is an unmistakable and tangible distinction between the water samples, as can be seen from the formation of effective clusters for MW and TW. Regardless, since the clustering observed for UW was not as successful as for the other waters, not even displaying a sequence or an order, additional PCA graphs, Figure 6, were plotted with only the calculated data from MW and TW, to go into more detail regarding these two more responsive media. It becomes apparent that the sensors are not only being able to discriminate between each type of water, in the case of TW an outstanding separation (and greater than perceived for MW) can be observed that follows an ordered sequence from the lowest to the highest concentration along the Principal Component 2′s axis. It is also relevant to note that sensors’ sensitivity is higher at lower concentrations (10^−15^ to 10^−12^ M), and as the concentration is seen to increase, it seems to build up to a saturation point (close to 10^−9^ M). Moreover, when Principal Component 2 (PC2) values from the data of Figure 6 are plotted as a function of concentration in logarithm scale, Figure 7a–c are achieved. These figures reveal that PC2 data achieved from the capacitance and loss tangent spectra measured in TW follow the EE2 concentration, while PC2 data calculated from the electric modulus spectra measured in MW follow the EE2 concentration.

In addition, considering a small range of EE2 concentrations, i.e., from 10^−14^ to 10^−10^ M, and by plotting the Principal Component 2′s axis for both waters as a function of the concentration of EE2 in the same graph, as shown in Figure 8, one can observe a constant proportionality between the PC2 values achieved for the two types of water attained at each EE2 concentration. This proportionality between the PC2 values for all the EE2 concentrations reveals that one can use this methodology to quantify EE2 concentration in unknown aqueous complex matrices.

Notwithstanding the fact that both MW curves display different behaviors and do not exhibit a linear trend, Figure 7a,b, it is possible to verify the opposite for TW given that it presents a more linear tendency, which is a far more complex matrix than MW, and, in turn, directly impacts the sensitivity of the sensors in their ability to detect, separate and discriminate between the concentrations analyzed. These data allow us to calculate the sensor sensitivity. Values of −0.072 ± 0.009 and −0.44 ± 0.03 per decade of concentration are achieved for the sensor system sensitivity with respect to EE2 in mineral and tap water, respectively, as well as detection limits of 7.5 × 10^−15^ M (22.2 pg/L) and 8.6 × 10^−15^ M (2.6 pg/L) in MW and TW, respectively. To better understand the obtained results, a comparison of this sensor system with the other EE2 sensors of referred to in the literature was conducted. Table 1 summarizes the results achieved by the different type of EE2 sensors present in the literature and different measurement techniques.

From the results displayed in this table, one can state that the type of sensor proposed in this work presents a lower detection limit when compared to the other sensors, which reached limits as low as pg/L. This confirms that these sensors, as well as the measurement technique, should be further used to develop an optimized EE2 sensor.

## 5. Conclusions

In this work, an EE2 sensor device, which allows the detection of the hormone concentration below 10^−9^ M in complex aqueous media by measuring the loss tangent spectra, in described. This device was based on two solid supports with deposited gold IDE, one without any thin film and the other with a (PEI/PSS)_5_ thin film deposited onto the IDE. Capacitance, loss tangent and electric modulus spectra were measured to characterize this sensor, and data were analyzed by the PCA method. Data analysis revealed that all capacitance, loss tangent and electric modulus spectra can discriminate the aqueous media, and the EE2 concentrations in the case of tap water. Moreover, the loss tangent spectra data can discriminate the EE2 concentrations in both mineral and tap water, and the achieved response values follow the same behavior as the EE2 concentration in these two, complex media. This allowed the calculation of the sensors’ a) sensitivity values of −0.072 ± 0.009 and −0.44 ± 0.03 per decade of concentration, for mineral and tap water, respectively, and b) detection limits values of 8.6 fM (2.6 pg/L) and of 7.5 fM (22.2 pg/L) for tap and mineral waters, respectively. These last values were found to be lower than the ones found in the literature for EE2 sensors.

This sensor system also revealed an increasingly better response as the complexity of the water under scrutiny increases, which may lead to promising future studies with wastewater or even river water samples. It was also possible to observe that the IDE sensors with the (PEI/PSS)_5_ thin film coating show a better sensitivity when exposed to the same conditions and samples as their uncoated (no film) counterpart. This is explained by the possible electrochemical reactions which can occur when the IDE is uncoated. Through the different studies and results obtained here, one can conclude that the developed IDE sensor-based systems are able to classify and discriminate between different water types, distinguish differing concentrations and demonstrate an ordered sequence of concentrations. One expects that an array of sensors with some of the units having specificity to EE2 will allow the development of a sensitive, cheap, and easy-to-handle sensor device, capable of detecting this hormone in complex aqueous matrices.

## Figures and Tables

**Figure 1 sensors-20-07324-f001:**
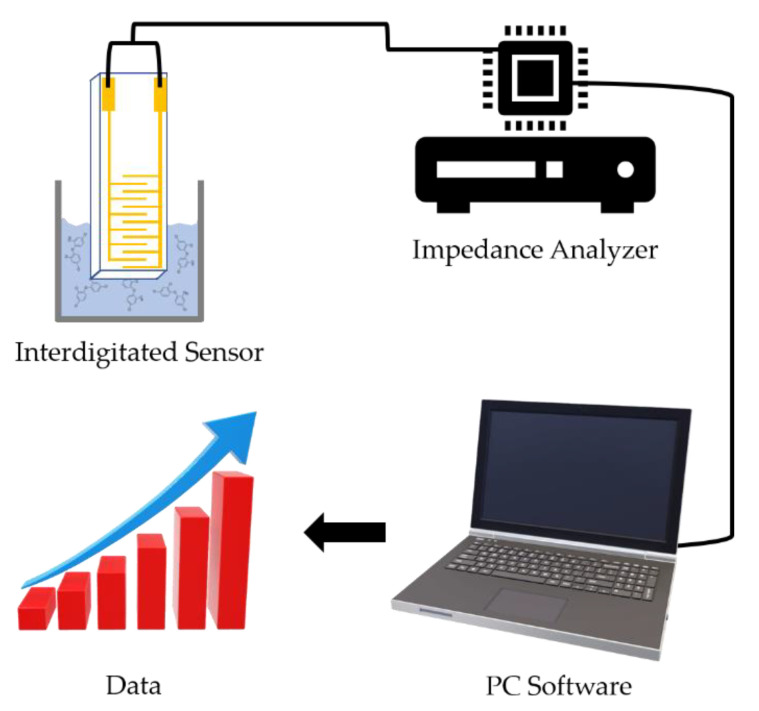
Schematic illustration of the experimental setup.

**Figure 2 sensors-20-07324-f002:**
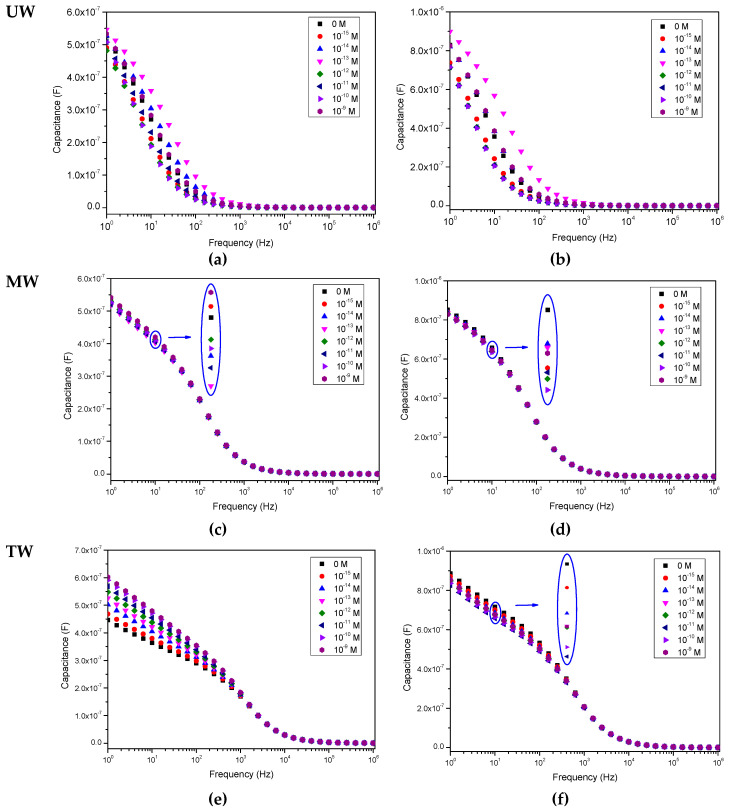
Capacitance spectra of the sensor devices when immersed in water solutions spiked with different EE2 concentrations: (**a**) uncoated interdigitated electrodes (IDE) in ultrapure water (UW), (**b**) IDE coated with (PEI/PSS)_5_ film in UW, (**c**) uncoated IDE in mineral water (MW), (**d**) IDE coated with (PEI/PSS)_5_ film in MW, (**e**) uncoated IDE in TW, (**f**) IDE coated with (PEI/PSS)_5_ film in tap water (TW). Relative error is less than 1%.

**Figure 3 sensors-20-07324-f003:**
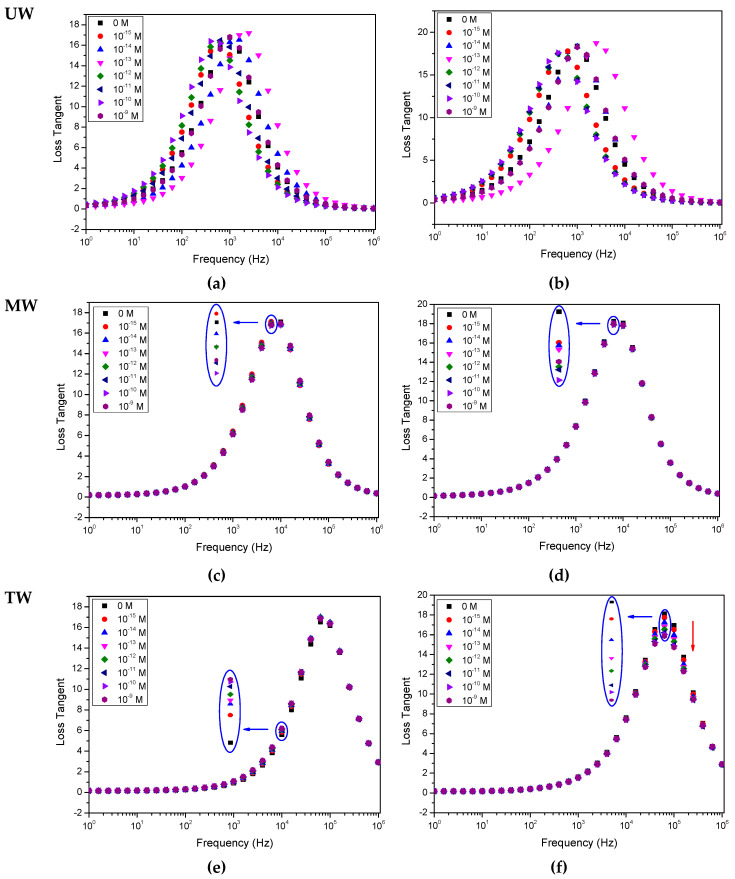
Loss tangent spectra of the sensor devices when immersed in water solutions spiked with different EE2 concentrations: (**a**) uncoated IDE in UW, (**b**) IDE coated with (PEI/PSS)_5_ film in UW, (**c**) uncoated IDE in MW, (**d**) IDE coated with (PEI/PSS)_5_ film in MW, (**e**) uncoated IDE in TW, (**f**) IDE coated with (PEI/PSS)_5_ film in TW. Relative error is less than 1%.

**Figure 4 sensors-20-07324-f004:**
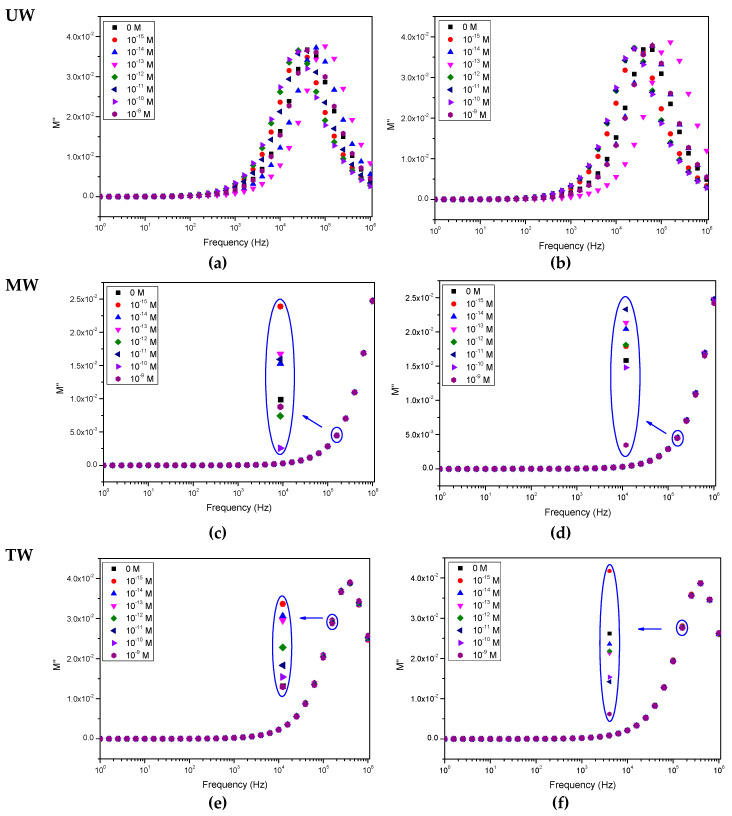
Spectra of the imaginary part of the electrical modulus (M’’) of the sensor devices when immersed in water solutions spiked with different EE2 concentrations: (**a**) uncoated IDE in UW, (**b**) IDE coated with (PEI/PSS)_5_ film in UW, (**c**) uncoated IDE in MW, (**d**) IDE coated with (PEI/PSS)_5_ film in MW, (**e**) uncoated IDE in TW, (**f**) IDE coated with (PEI/PSS)_5_ film in TW. Relative error is less than 1%.

**Figure 5 sensors-20-07324-f005:**
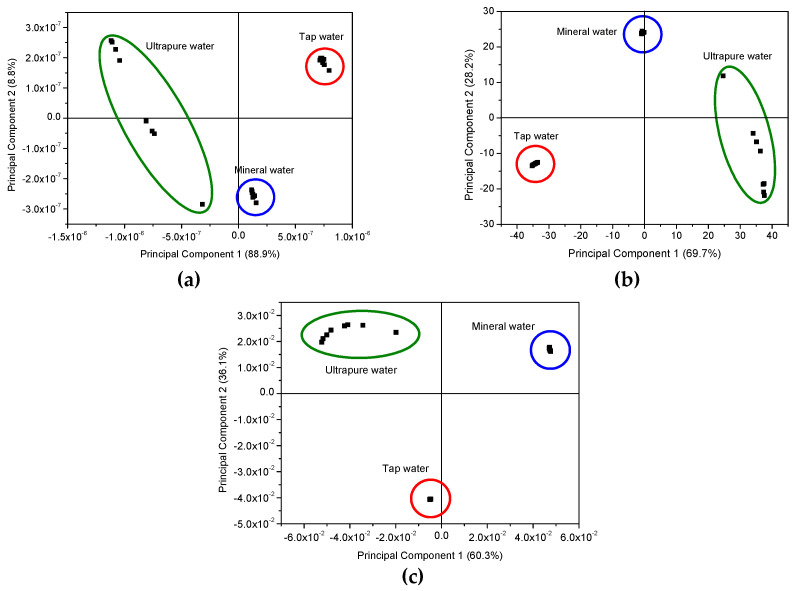
PCA plots using the capacitance (**a**), loss tangent (**b**,**c**) electric modulus spectra measured when both uncoated IDE and coated IDE with (PEI/PSS)_5_ film are immersed in spiked solutions with EE2 concentrations (0 to 10^−9^ M) prepared with the three types of water used. The colored circular and oval lines surround the PCA achieved data associated with the different waters.

**Figure 6 sensors-20-07324-f006:**
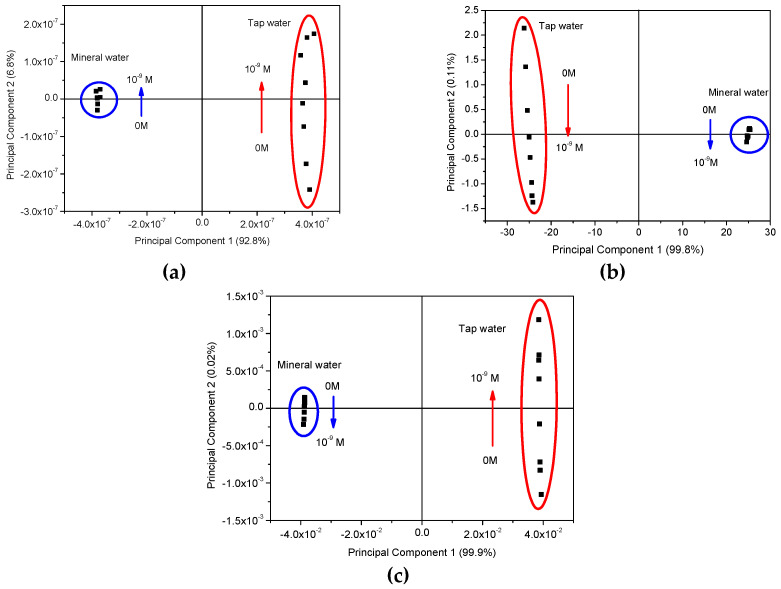
PCA plots using the capacitance (**a**), loss tangent (**b**,**c**) electric modulus spectra measured when both uncoated IDE and coated IDE with (PEI/PSS)_5_ film are immersed in spiked solutions with EE2 concentrations (10^−9^ M to 0 M) prepared with) mineral water and tap water. The colored circular and oval lines surround the PCA achieved data associated with the different waters.

**Figure 7 sensors-20-07324-f007:**
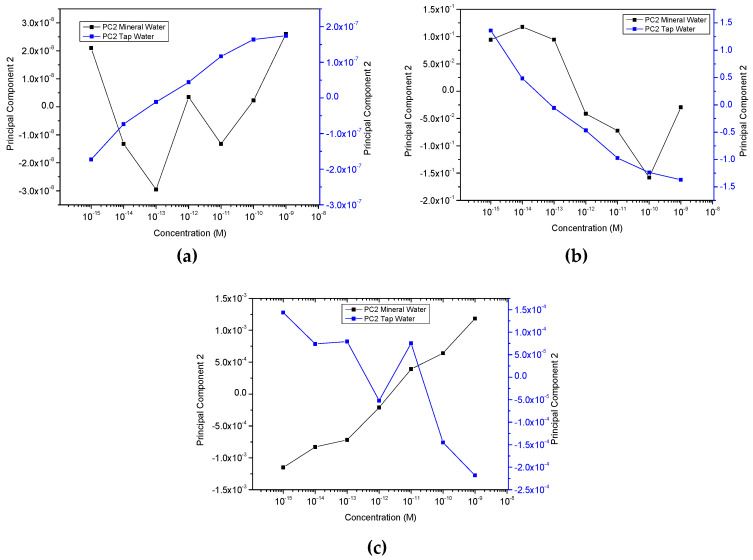
Plots of Principal Component 2 calculated values from capacitance (**a**), loss tangent (**b**,**c**) electric modulus spectra for mineral and tap waters as a function of logarithm of the sample’s concentrations. The lines between the plotted points are only guidelines.

**Figure 8 sensors-20-07324-f008:**
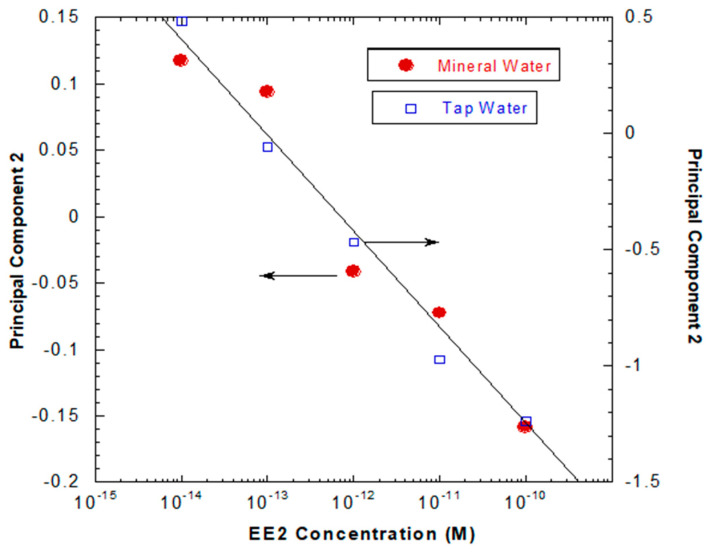
Plot illustrating the constant proportionality achieved when the sensor’s responses attained in both mineral and tap waters are plotted as a function of EE2 concentration.

**Table 1 sensors-20-07324-t001:** Comparison of the responses of sensors in the literature with those of this work.

Detection Technique	Sensor	Detection Limit
Diff. Pulse Voltammetry [15]	Au/Fe3O4@TA/MWNT/GCE	3.3 nM
Electrochemical Detection [16]	PVP/Chi/rGO_Laccase	0.15 pmol/L
Adsorptive Stripping Voltammetry [17]	HDME	14.8 μg/L at −0.23 V
	HDME	9.7 μg/L at −1.20 V
	SPCE	182 μg/L
	SPCNTE	191 μg/L
Sq. Wave Voltammetry [18]	LbL FTO-(Chi/CNT)3	0.009 μmol/L
Impedance Spectroscopy (this work)	Uncoated IDE and IDE coated with (PEI/PSS)_5_ film (from 0 M to 10^−9^ M)	7.5 fM (22.2 pg/L) for Mineral Water 8.6 fM (2.6 pg/L) for Tap Water

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
