# Peer review of "Detecting Traces of 17α-Ethinylestradiol in Complex Water Matrices"

_sensors, 2020, doi:10.3390/s20247324_

Round 1

Reviewer 1 Report

The dielectric spectroscopy measurements are likely to suffer from undesirable electrode polarization effects, which, in turn, affect the relxataionmechanisms studied. I suggest that the authors should provide additional data maniulation by emplying the electric modulus function M*. Its imaginary part im(M*) supress - by definition electrode polarization.

The author should revise their manuscript toadress these problem.

Author Response

Resp: We appreciate the reviewer's comment which makes perfect sense. Therefore we revised the manuscript as suggested. We know that the electric modulus is quite an interesting physical quantity. So we decided to add these spectra to the paper for comparison with other functions. In fact, this physical quantity is also associated to protonic conduction which can occur in our case. The use of the imaginary part of the electric modulus was successful in the determination of the evolution of PC2 with EE2 concentration in the mineral matrix, which is an aspect that should be considered in the development of this type of sensors. Some references were also included in the manuscript.

Reviewer 2 Report

Line 70 – Working electrode.

Could rephrase the entire paragraph (Line 66 to 71) to sound more technically sound.

Line 74 – When using words like “better” please have a comparison. Better than what? Or else please be more specific with the choice of words.

Line 78 – Needs to be rephrased.

Line 79 – 83 – Lots of grammatical errors. Please Rephrase

Line 92 to 95 – Too long for a sentence. Please break this sentence.

Line 103 to 105 – Please explain the abbreviated words like PC2, MW and TW on the first instance of usage in the document.

Line 108 – What representation of dimension is 200um/200um? Please be specific if it is the length and width or length and the thickness.

Figure 1 Is not readable. Please use a high-resolution figure format.

Also Figure 1 where the author compares the capacitance change with increasing EE2 concentration, there is no clear trend on how the capacitance change correlates with the increasing concentration. I see the shift in the Capacitance trend is random with linear increase in concentration.

Also, the author argues that the capacitance and loss tangent is coherent in fig 1f and 2f. But I see a much larger sensitivity (Change in capacitance) and coherence in figure 1e. The authors need to provide an explanation as to why this happens. Also, what advantage does the PEI/PSS coating that they claim to improve sensitivity have over the naked uncoated IDE’s? Basically, Paragraph (159 to 166) needs to have more supportive arguments rather than describing the plot results.

The author claims that Figure 1 and 2 produced repeatable data. The authors need to show the reproducibility of some of the capacitance and loss tangent values with error bars to claim reproducibility.

Is the description of (a) and (b) in figure 4 right? It does not make sense. Is (a) the uncoated IDE and (b) the coated IDE?

Line 198 – What is “intense change”??? Please avoid using such words and be more specific while discussing results.

Except figure 2, all other figures are not readable!!

I agree there is a good linear trend for Tap water. There is no linear trend for MW in both cases in figure 5!!

The author claims this sensor is able to distinguish between different kinds of water looking at the shift in curve towards right in Fig 1 and 2. I do not see how this sensor system can be claimed to be an efficient and simple sensor to detect EE2 levels. How complex are the post processing steps once you obtain the data? It is not a simple interpolation or averaging of the raw signal in the authors sensor.

There is no figures or optical illustration of the sensor or the experimental setup used. Although it might be referring to a previously developed work, the authors need to portray the basic geometry of the sensor setup used to the reader.

Overall, the introduction and background is good with some need for grammatical correction. The figures are unreadable. There needs to be better scientific interpretation of the plots and not just stating the obvious trend in a graph. There needs to be strong arguments placed and justifications for all the claims made.

Author Response

Reviewer2

Line 70 – Working electrode.

Could rephrase the entire paragraph (Line 66 to 71) to sound more technically sound.

Resp: The paragraph was rephrased. Thank you very much.

Line 74 – When using words like “better” please have a comparison. Better than what? Or else please be more specific with the choice of words.

Resp: The sentence was improved.

Line 78 – Needs to be rephrased.

Resp: The sentence was improved.

Line 79 – 83 – Lots of grammatical errors. Please Rephrase

Resp: The text was improved.

Line 92 to 95 – Too long for a sentence. Please break this sentence.

Resp: The sentence was broken.

Line 103 to 105 – Please explain the abbreviated words like PC2, MW and TW on the first instance of usage in the document.

Resp: The abbreviated words were explained.

Line 108 – What representation of dimension is 200um/200um? Please be specific if it is the length and width or length and the thickness.

Resp: The characteristics of the electrodes were better explained in the manuscript.

Figure 1 Is not readable. Please use a high-resolution figure format.

Resp: We apologize for the inconvenience of the figures not being legible. Something unbeknownst to us must have happened, since in our version all figures are legible and have quality.

Also Figure 1 where the author compares the capacitance change with increasing EE2 concentration, there is no clear trend on how the capacitance change correlates with the increasing concentration. I see the shift in the Capacitance trend is random with linear increase in concentration.

Resp: The reviewer is right. Some text was missing. Now, some sentences were added to the manuscript.

Also, the author argues that the capacitance and loss tangent is coherent in fig 1f and 2f. But I see a much larger sensitivity (Change in capacitance) and coherence in figure 1e. The authors need to provide an explanation as to why this happens. Also, what advantage does the PEI/PSS coating that they claim to improve sensitivity have over the naked uncoated IDE’s? Basically, Paragraph (159 to 166) needs to have more supportive arguments rather than describing the plot results.

Resp: Again the reviewer is right. An explanation was added. The paragraph was changed for better understanding.

The author claims that Figure 1 and 2 produced repeatable data. The authors need to show the reproducibility of some of the capacitance and loss tangent values with error bars to claim reproducibility.

Resp: The information about the error bars was added in the text and in the figure captions.

Is the description of (a) and (b) in figure 4 right? It does not make sense. Is (a) the uncoated IDE and (b) the coated IDE?

Resp: The data used in these PCA plots encompasses both coated and uncoated IDE. (a) and (b) are referring to which spectra data was used to plot the PCA graphs, which in this case were capacitance and loss tangent, respectively. The figure captions text was improved to be clearer in the manuscript.

Line 198 – What is “intense change”??? Please avoid using such words and be more specific while discussing results.

Resp: The sentence was improved. Thank you.

Except figure 2, all other figures are not readable!!

Resp: We apologize for the inconvenience of the figures not being legible. Something unbeknownst to us must have happened, since in our version all figures are legible and have quality.

I agree there is a good linear trend for Tap water. There is no linear trend for MW in both cases in figure 5!!

Resp: Yes, it is correct. But if we consider a small range of concentration is possible to find a linear region. This information is included in the manuscript.

The author claims this sensor is able to distinguish between different kinds of water looking at the shift in curve towards right in Fig 1 and 2. I do not see how this sensor system can be claimed to be an efficient and simple sensor to detect EE2 levels. How complex are the post processing steps once you obtain the data? It is not a simple interpolation or averaging of the raw signal in the authors sensor.

Resp: In fact, the spectra of the three physical quantities are different and dependent of the aqueous solution, therefore, impedance spectra distinguish the aqueous media. We analyze the data by applying directly the PCA to the different type of spectra independently. After that we analyze the principal component 2 with the EE2 concentration. This is due to the water matrix having a strong effect on the principal component (1), given that each type of aqueous matrix presents different spectra.                

There is no figures or optical illustration of the sensor or the experimental setup used. Although it might be referring to a previously developed work, the authors need to portray the basic geometry of the sensor setup used to the reader.

Resp: A schematic representation of the measuring system was included, now figure 1.

Overall, the introduction and background is good with some need for grammatical correction. The figures are unreadable. There needs to be better scientific interpretation of the plots and not just stating the obvious trend in a graph. There needs to be strong arguments placed and justifications for all the claims made.

Resp: Thank you very much for your comments that contribute to the clarification and improvement of the manuscript.

Reviewer 3 Report

  1. Format issues:
  • Modify all the unclear diagrams, which will greatly affect the readers' reading experience.
  • Form using 3-line table.
  • Milliq water and ultrapur water should be unified in the paper and diagram.
  • Figures are not clear, except Fig. 2.
  1. Add a device diagram to the Materials and Methods to illustrate the test method.
  2. PCA is a good idea to distinguish kinds and concentration information, but I think the PCA dimensionality reduction you do is not universal and generalized. Please optimize the PCA model in the revised manuscript to achieve better results. The reasons are as follows:
  • Figure 4 is the result obtained by removing the ultrapur water data in figure 3, and the reason for this result can be found in figure 2. The data difference between different concentrations in ultrapur water is too large, which inundates the concentration information of mineral water and tap water.
  • And in your conclusion you also mentioned "This sensor system also revealed an increasingly better response as the complexity of the water under scrutiny increases".You put forward this prediction because the more complex the water quality environment, the greater the difference between concentrations.
  • When the sensor is in the complex water quality, it is likely that the data concentration information of good water quality will be flooded, so I think the PCA dimensionality reduction you do is not universal and generalized.
  1. As can be seen from Fig .2, IDE sensors coated with (PEI/PSS)5 thin films has better effect, and the more complex the water quality, the more obvious the concentration difference is. Please explain the causes of these two phenomena in the revised manuscript.
  2. How much data is used in the conclusions of figures 5 and 6? If only the data points in the graph, there is obviously contingency error.

Author Response

Reviewer3

Modify all the unclear diagrams, which will greatly affect the readers' reading experience.

Resp: We apologize for the inconvenience of the figures not being legible. Something unbeknownst to us must have, since in our version all figures are legible and have quality.

Form using 3-line table.

Resp: The table was improved.

MilliQ water and ultrapure water should be unified in the paper and diagram.

Resp: Now we use ultrapure water in the manuscript. Thank you.

Figures are not clear, except Fig. 2.

Resp: We apologize for the inconvenience of the figures not being legible. Something unbeknownst to us must have, since in our version all figures are legible and have quality.

Add a device diagram to the Materials and Methods to illustrate the test method.

Resp: A schematic representation of the measuring system was included, now figure 1.

PCA is a good idea to distinguish kinds and concentration information, but I think the PCA dimensionality reduction you do is not universal and generalized. Please optimize the PCA model in the revised manuscript to achieve better results. The reasons are as follows:

- Figure 4 is the result obtained by removing the ultrapure water data in figure 3, and the reason for this result can be found in figure 2. The data difference between different concentrations in ultrapure water is too large, which inundates the concentration information of mineral water and tap water.

- And in your conclusion you also mentioned "This sensor system also revealed an increasingly better response as the complexity of the water under scrutiny increases". You put forward this prediction because the more complex the water quality environment, the greater the difference between concentrations.

- When the sensor is in the complex water quality, it is likely that the data concentration information of good water quality will be flooded, so I think the PCA dimensionality reduction you do is not universal and generalized.

Resp: Thank you very much for the comment because it is of extreme importance. Bearing in mind that the electrical measurements in the ultrapure aqueous matrix are somewhat random, we applied the PCA method only to samples prepared in MW and TW. This was done in order to remove the error inserted by UW. However, we are aware that the contingency error can be high but it is quite difficult to measure a larger number of samples. We added this information in the manuscript. Moreover, if we find good sensors that can distinguish the different concentrations, use the adequate type of measuring spectra and a large library of calibrated spectra, definitely, the PCA method will give excellent results.

As can be seen from Fig .2, IDE sensors coated with (PEI/PSS)5 thin films has better effect, and the more complex the water quality, the more obvious the concentration difference is. Please explain the causes of these two phenomena in the revised manuscript.

Resp:. The presence of a thin film coating on the IDE prevent certain electrochemical reactions that can occur during the electric signal measurement. In certain cases, with increasing measuring times, these reactions can lead to a partial removal of the gold electrodes. Also, the presence of a negative polyelectrolyte layer on the sensor’s surface does not allow EE2 or other negative molecules or negative ions to be adsorbed on the surface. This creates a new dynamic on the electric double layer, a distribution in the space of negative and positive ions that appears when a surface is exposed in the aqueous solution and allows the sensor to be used several times. Finally, the more complex a water matrix, more electrical charges are at play due to the presence of a larger amount of ions as well as a larger amount molecules, which allows higher electrical conduction. It should also be referred that in the case of ultrapure water matrix, the amount of ions is reduced thus becoming difficult for the electric double layer to be formed when the sensor is immersed in the water. The random changes present in the impedance spectra when the aqueous matrix is ultrapure water can be explained by the formation of electric double layers which display a random nature due to a presence of random number of positive and negative of ions. Concerning the (PEI/PSS)5 thin film, PEI (positively charged) and PSS (negatively charged) are both polyelectrolytes and also insulators. Its presence works as capacitor, influencing the features of the impedance spectra measured. These phenomena were now better explained in the manuscript.

How much data is used in the conclusions of figures 5 and 6? If only the data points in the graph, there is obviously contingency error.

Thank you very much for the comment.  In fact, the PCA should be used in a large number of samples using a large number of variables, see for example [Osborne, Jason W. & Anna B. Costello (2004). Sample size and subject to item ratio in principal components analysis. Practical Assessment, Research & Evaluation, 9(11). Available online: http://PAREonline.net/getvn.asp?v=9&n=11.] which is in accordance with the main concept of electronic tongue where an array of sensors is used to find a large number of variables and compare the measured data with the data of a large number of samples existing in a library. However, from the point of view of sensor development, in which it is necessary to study different proprieties, it becomes impracticable to obtain as many samples as necessary to reduce the error. Here we intend to show the proof of concept that the choice of physical quantities may have an importance in the sensor response as well as these sensors allow quantification in addition to being qualitative. Therefore, it was included a sentence that clarifies the ratio between samples and variables used in this work.

Round 2

Reviewer 1 Report

The authors have improved their manuscript substantially.

 It is necessary to update recent progress and citing in using the electric modulus  formalism:

Page 3, line 99: Electric modulus is usually related to proton conduction and, recently, linked to electron percolation through tunneling in polymer based composites [Materials Chemistry and Physics  232, 319-324 (2019); Mat. Chem, Phys. 2, 140 (2019); J. Phys. D: Appl. Phys. 48, 285305   (2016)].

Reviewer 2 Report

Thank you for implementing the suggested changes. 

Please change the y-scale to nanoFarads or microFarads for the capacitance values. Mention the units of Capacitance measured. 

Reviewer 3 Report

The changes to the manuscript meet the reviewers’ requests. So, I recommend the manuscript be accepted for publication.